# Tuning colloidal quantum dot band edge positions through solution-phase surface chemistry modification

Daniel M. Kroupa[1,2,*], Márton Vörös[3,4,*], Nicholas P. Brawand[4], Brett W. McNichols[5], Elisa M. Miller[1], Jing Gu[1], Arthur J. Nozik[1,2], Alan Sellinger[1,5], Giulia Galli[3,4] & Matthew C. Beard[1]

Band edge positions of semiconductors determine their functionality in many optoelectronic applications such as photovoltaics, photoelectrochemical cells and light emitting diodes. Here we show that band edge positions of lead sulfide (PbS) colloidal semiconductor nanocrystals, specifically quantum dots (QDs), can be tuned over 2.0 eV through surface chemistry modification. We achieved this remarkable control through the development of simple, robust and scalable solution-phase ligand exchange methods, which completely replace native ligands with functionalized cinnamate ligands, allowing for well-defined, highly tunable chemical systems. By combining experiments and *ab initio* simulations, we establish clear relationships between QD surface chemistry and the band edge positions of ligand/QD hybrid systems. We find that in addition to ligand dipole, inter-QD ligand shell inter-digitization contributes to the band edge shifts. We expect that our established relationships and principles can help guide future optimization of functional organic/inorganic hybrid nanostructures for diverse optoelectronic applications.

[1] Chemistry & Nanoscience Center, National Renewable Energy Laboratory, Golden, Colorado 80401, USA. [2] Department of Chemistry and Biochemistry, University of Colorado, Boulder, Colorado 80309, USA. [3] Materials Science Division, Argonne National Laboratory, Lemont, Illinois 60439, USA. [4] Institute for Molecular Engineering, University of Chicago, Chicago, Illinois 60637, USA. [5] Department of Chemistry and Materials Science Program, Colorado School of Mines, Golden, Colorado 80401, USA. * These authors contributed equally to this work. Correspondence and requests for materials should be addressed to A.S. (email: aselli@mines.edu) or to G.G. (email: gagalli@uchicago.edu) or to M.C.B. (email: matt.beard@nrel.gov).

Colloidal semiconductor nanocrystals, specifically quantum dots (QDs), are of interest to numerous scientific disciplines due to their highly tunable optical and electronic properties. The study of QDs has long been focused on the inorganic core; specifically, on quantum-confinement (for example, size-dependent band gaps[1] and enhanced Auger-type processes[2]) or increased surface-to-volume ratio effects (for example, size-dependent phase transitions[3]). However, it has become increasingly clear that post-synthetic surface chemistry modification, or ligand exchange, can critically influence QD optoelectronic properties, as well[4–7]. Ligand exchanges are often performed in the solid state, where films of QDs with long, aliphatic surface ligands are exposed to solutions of shorter alkyl-chain or atomic ligands for exchange. These types of ligand exchanges are convenient methods for constructing functional materials; however, establishing fundamental principles that govern the relationship between ligand/QD optoelectronic properties and the physiochemical nature of the surface is not straightforward, as many uncontrolled and ill-defined variables are introduced during QD film processing, including the extent of ligand exchange, QD core stoichiometry, facet-specific ligand coordination and ligand denticity.

For the case of metal chalcogenide QDs, several research groups have studied the influence of surface chemistry on QD optoelectronic properties. Surface chemistry modification has long been used to effectively passivate surface states and electronically couple QDs through decreased inter-QD distance, thus leading to enhanced carrier transport in QD thin films and improved photoluminescence quantum yields[8,9]. More recently, researchers have studied how ligand/QD optoelectronic properties are influenced by ligand functionalization through the variation of ligand coordination environment and/or ligand electron donating/withdrawing character[10–13]. The Bawendi[14,15], Luther[16] and Bent[17] groups found that surface chemistry modification using solid-state ligand exchange techniques can shift the ionization energy (IE) and work function ($\Phi$) of lead sulfide (PbS) QD thin films, thus allowing for the engineering of more efficient QD solar cells. All of these studies demonstrated that modifying the ligand/PbS QD interface produces quite distinct chemical systems, and some even suggested a link between QD band edge energy shifts and ligand dipole moment; however, due to the uncontrolled and ill-defined physiochemical nature of solid-state ligand exchanges, a clear and quantitative relationship has never been reported.

To this extent, we have developed a simple, robust and scalable solution-phase X-type ligand exchange method for PbS QDs that replaces native surface ligands with functionalized cinnamate ligands, yielding highly tunable, well-defined organic/inorganic hybrid chemical systems. We explore a library of functionalized cinnamic acid molecules to systematically tune PbS QD surface chemistry, and find that thin films of fully ligand-exchanged QDs exhibit remarkable band edge shifts: the band edge position of QDs can be tuned over 2.0 eV, the largest value reported to date. We use a combination of experiment and theory to directly establish the physical principles governing QD surface chemistry-induced band edge shifting, and find that ligand dipoles alone are insufficient to fully describe the observed band edge shifts. We propose that inter-QD ligand shell inter-digitization likely present in close packed QD thin films must be accounted for, and we report quantitative comparisons between theory and experiment.

## Results

**Solution-phase surface chemistry modification.** We identified seven functionalized cinnamic acid (R-CAH) ligands (Fig. 1a) as candidates for solution-phase ligand exchanges at the oleate (OA$^-$) terminated, lead-rich surface of 3.2 nm diameter PbS QDs (QD synthesis, adapted from reference[18], is detailed in Methods section). R-CAH molecules are ideal ligands for solution-phase exchange: the optoelectronic properties of the R-CAHs are widely tunable through functionalization of the aromatic ring of the ligand motif; the vinyl linkage of R-CAH allows for electronic coupling of the dipole active portion of the ligand to the QD core; the R-CA$^-$ ligands impart long-term colloidal stability and prevents QD aggregation; the carboxylate surface coordination environment of native oleate (OA$^-$) is conserved post-exchange; and finally, the binding of R-CAHs induces broadband optical absorbance enhancement, which can be utilized to monitor the extent of exchange *in situ*[10,11,19]. We classify each R-CAH ligand by its functional group and whether it is protonated and free in solution or deprotonated and bound to the QD surface. Thus, 4-H-CAH denotes *trans*-cinnamic acid and 4-H-CA$^-$ denotes *trans*-cinnamate. Of the ligands studied, only 4-CN-CAH was not commercially available and was synthesized in house (details in Supplementary Note 1). We found the calculated vacuum electronic dipole moment of the unbound, protonated R-CAHs to be readily tunable through functionalization of the aromatic ring of the cinnamic acid ligand motif (Fig. 1a, computational details can be found in Methods section and Supplementary Note 2).

We developed a simple, robust and scalable solution-phase ligand exchange method for the R-CAH ligands of interest in this work following the scheme described in Fig. 1b. Purification of the fully exchanged PbS QDs involved removal of the oleic acid (OAH) byproduct and residual R-CAH through multiple cycles of selective precipitation of the ligand-exchanged QDs from solution using a non-polar antisolvent (hexane), followed by centrifugation and redissolution in an appropriate solvent (PCR purification; ligand exchange details can be found in Methods section and Supplementary Table 1).

A combination of Fourier transform infrared spectroscopy (FTIR; Fig. 2a) and $^1$H nuclear magnetic resonance spectroscopy (NMR; Fig. 2b) provided direct evidence that OA$^-$ is removed from the QD surface via an exchange with R-CA$^-$. The FTIR spectrum of OAH has a broad –OH feature from 2,250–3,250 cm$^{-1}$, alkane/alkene C-H stretches around 3,000 cm$^{-1}$ and a distinct C=O stretch near 1,680 cm$^{-1}$, while for the OA$^-$/QD complex, the alkane/alkene C–H stretches are retained, but the broad –OH and sharp C=O stretches are no longer present. Instead, evidence of a bidentate carboxylate binding environment is observed with symmetric and asymmetric COO$^-$ stretches at 1,530 and 1,408 cm$^{-1}$. For each ligand exchange, we found comparable spectral changes between free R-CAHs and R-CA$^-$/QDs, suggesting that the R-CA$^-$ ligands coordinate the QD surface in a similar geometry as OA$^-$ and no free R-CAH remains. In addition, comparing the OA$^-$/QD spectrum to the R-CA$^-$/QD spectra, we observed a drastic decrease in the alkane/alkene C–H stretches (~3,000 cm$^{-1}$), indicating efficient displacement of OA$^-$ and removal of OAH. In addition, the unique C≡N triple bond stretching can be clearly observed near 2,250 cm$^{-1}$ for 4-CN-CA$^-$/QD and 4-CN-CAH.

Furthermore, a comparison of the $^1$H NMR spectra (Fig. 2b) of free 4-H-CAH and 4-H-CA$^-$/QD complex after purification showed distinct differences. The vinyl peak at 5.3 p.p.m. is significantly broadened for the OA$^-$/QD complex and shifted upfield due to dipolar coupling[20]. The OA$^-$/QD spectrum showed a lack of the broad OAH acidic proton peak around 12 p.p.m., in agreement with OA$^-$ chemical identity. Pure 4-H-CAH showed a doublet at 6.47 and 7.81 p.p.m. from the alpha and beta vinyl protons, respectively, with a broad peak

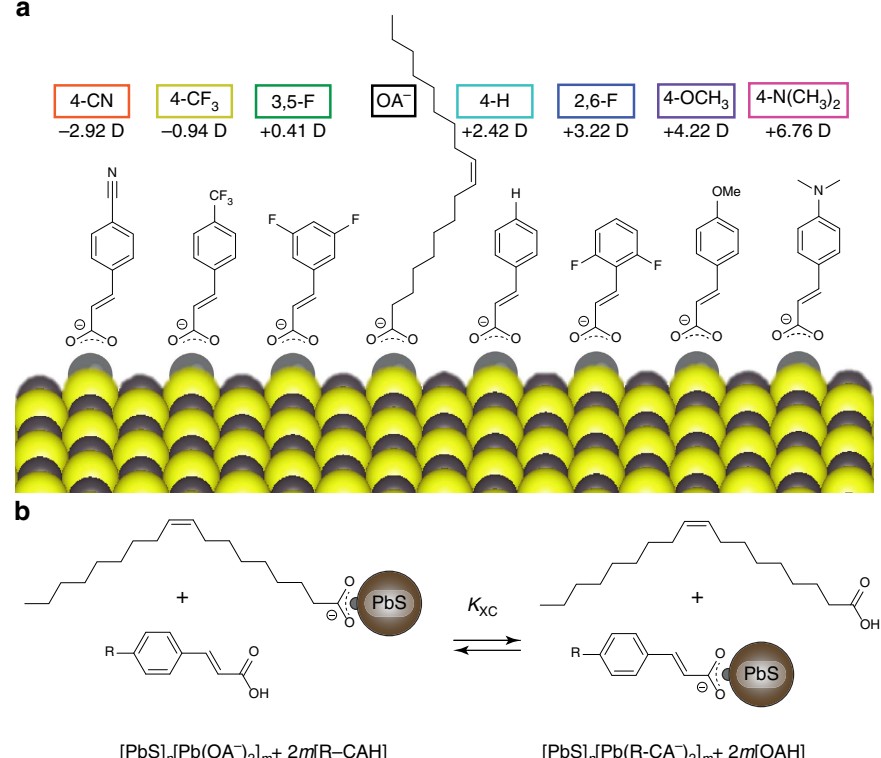

**Figure 1 | The model ligand/QD system utilized in this study. (a)** Chemical structures, computed vacuum electronic dipoles and labels used throughout this work of the molecules in our ligand library. 4-CN-CA$^-$ = 4-cyanocinnamate; 4-CF$_3$-CA$^-$ = 4-trifluoromethylcinnamate; 3,5-F-CA$^-$ = 3,5-difluorocinnamate; 4-H-CA$^-$ = cinnamate; 2,6-F-CA$^-$ = 2,6-difluorocinnamate; 4-OCH$_3$-CA$^-$ = 4-methoxycinnamate; 4-N(CH$_3$)$_2$-CA$^-$ = 4-dimethylaminocinnamate. OA$^-$ = oleate. **(b)** The X-type ligand exchange in which surface bound oleate is displaced by functionalized cinnamic acid molecules.

around 11.8 p.p.m. from the acidic proton. The remaining peaks at 7.56, 7.40 and 7.42 p.p.m. correspond to *ortho*, *meta* and *para* aromatic protons, respectively. The 4-H-CA$^-$/QD spectrum showed drastically different features than those of the OA$^-$/QD complex and pure 4-H-CAH. The 4-H-CA$^-$/QD aromatic and vinyl protons shift upfield and broaden significantly, accompanied by a concurrent loss of the carboxylic acid peak. Finally, the lack of the broad surface bound OA$^-$ vinyl peak suggested there is very little residual OAH or OA$^-$ in the ligand-exchanged sample after PCR purification. The combination of FTIR and $^1$H NMR spectroscopic analysis suggests the efficient and complete exchange of native OA$^-$ ligands for R-CA$^-$ followed by removal of excess R-CAH and OAH through PCR purification.

**Band edge shifts of ligand-exchanged PbS QDs.** Using the library of completely exchanged R-CA$^-$/QDs, we cast films of QDs via a single-deposition step on Au/glass substrates for X-ray photoelectron spectroscopy (XPS) measurements to extract the work function and valence band maximum ($E_{VBM}$) with respect to the Fermi level ($E_F$); ($E_F - E_{VBM}$), which can be summed to yield the IE of the QD thin film. The raw XPS secondary electron cut-off and valence band region data sets are presented in Fig. 3a,b, respectively, and data analysis is discussed in Supplementary Note 3. We found that QD IE (Fig. 3c, solid rectangles) is shifted across the R-CA$^-$/QD library by 2.1 eV accompanied by a shift in $\Phi$ (Fig. 3c, black dashed) of 2.4 eV. As a guide, we also include the conduction band minimum ($E_{CBM}$, open rectangles) determined from summing the optical gap (opaque lines) and calculated exciton binding energy[14].

Each of the QD films exhibits a Fermi-level within the QD band gap indicative of n-type behaviour (closer to the conduction band) that is consistent with the roughly constant Pb:S ratio measured from XPS analysis (Supplementary Table 2)[21]. In accordance, the X-type ligand exchange does not involve the removal of surface Pb atoms. Note that for all of our ligand exchanges, the QDs are kept strictly air-free; once exposed to ambient conditions, we expect the degree of oxidation to vary significantly across the ligand set, affecting the resulting Fermi-level position within the QD band gap.

Our analysis of the experimental band edge shifting data suggests that ligands containing fluorinated functional groups behave differently from those that do not—a phenomenon that has been previously reported for solid-state surface modification of ITO and ZnO with functionalized phosphonate molecules[5]. In particular, the 4-CN-CA$^-$/QD complex has the most negative (electron withdrawing) projected ligand dipole, but its measured band edge is not as deep as either the 4-CF$_3$-CA$^-$/QD or 3,5-F-CA$^-$/QD complexes (Figs 3c and 4a). A linear least squares fit of the data that only includes the fluorinated ligand/QD complexes finds a best-fit slope of 0.45 eV per Debye (Fig. 4a, red dashed line). In stark contrast, a best-fit line through the data that excludes the fluorinated ligand/QD complexes finds a much shallower slope, 0.12 eV per Debye (Fig. 4a, blue dashed line). To confirm the generality of this observation, we tested a second class of ligands—functionalized benzenethiolates (4-R-S$^-$), namely 4-H-S$^-$ (Fig. 4a, blue open circle) and 4-CH$_3$-S$^-$ (Fig. 4a red open circle, solution-phase ligand exchange; green open circle, layer-by-layer ligand exchange). See Supplementary Note 4 for experimental details regarding the solution-phase ligand exchange using 4-CH$_3$-S$^-$ and

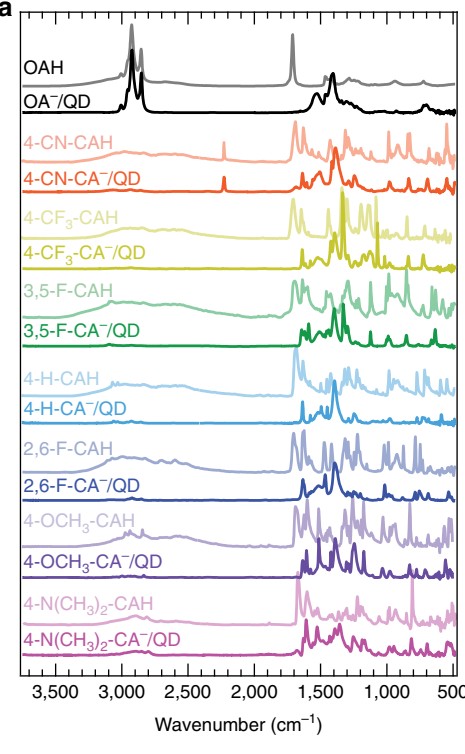

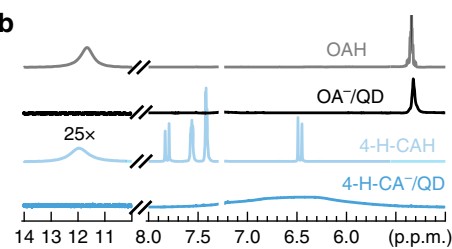

**Figure 2 | Surface analysis of PbS QDs before and after ligand exchange and PCR purification.** (**a**) FTIR spectra of neat ligand (lighter traces) and ligand/QD complex (darker traces) films. (**b**) $^1H$ NMR spectra in $CDCl_3$ of neat ligand (lighter traces) and ligand/QD complexes (darker traces). Both FTIR and $^1H$ NMR spectroscopic analysis suggests the efficient exchange of native $OA^-$ ligands for $R-CA^-$ and removal of excess $R-CAH$ and $OAH$ through PCR purification.

Supplementary Fig. 1 for FTIR spectra of the neat 4-$CH_3$-SH ligand and 4-$CH_3$-$S^-$/QD complex. Supplementary Fig. 2 and Supplementary Table 3 summarize the XPS data for the 4-H-$S^-$/QD complexes. We found good agreement between the measured absolute IE versus ligand dipole with that of the cinnamate ligands, suggesting that the surface dipole at the ligand/QD interface is also similar for both classes of ligands. To extend the benzenethiolate library, we plot the IE versus calculated 4-R-$S^-$ dipole taken from a separate data set collected by Bent et al.[17] (Fig. 4a, brown and grey open circles) using a similar PbS 4-R-$S^-$/QD material system but using different techniques to measure film IE (ambient PES) and calculate ligand dipole. We note that while their measured IE for a 4-$NO_2$-$S^-$/QD thin film (Fig. 4a, open brown circle) finds good agreement with our non-fluorinated ligand data set (Fig. 4a, blue dashed line), their measured IE for a 4-F-$S^-$/QD thin film (Fig. 4a, open grey circle) finds better agreement with our fluorinated ligand data set (Fig. 4a, red dashed line). Therefore, we conclude that the observation of fluorinated ligands inducing deeper band edge shifts compared with non-fluorinated ligands is a general trend.

To understand the magnitude and direction of the experimental band edge shift measurements and how they relate to cinnamic acid ligand functionalization, we carried out first principles calculations using three different structural models of ligand/QD complexes with varying surface coverage. Details of the ligand/QD structural models can be found in Supplementary Note 5, and ball-and-stick representations of the models can be found in the insert to Fig. 4b and Supplementary Fig. 3. Details of the R-$CA^-$/QD band edge energy calculations can be found in Supplementary Note 6. Our calculations showed that the band edge shifts, $\Delta E$, for the R-$CA^-$/QD complexes are proportional to the ligand dipole calculated in vacuum for a given ligand/QD model, and that the models with higher ligand surface coverage exhibit stronger dependence on the ligand dipole (Supplementary Fig. 4). Furthermore, the calculated $\Delta E$ renormalized by the ligand surface coverage for each ligand/QD model all show a similar, roughly linear trend with the calculated ligand dipole (Fig. 4b).

Since these ligand/QD complexes are smaller than those used in the experiments and are not fully ligated for computational reasons, we parameterized a simple electrostatic model[22] for better comparison with experiments. Assuming that the QD is spherical and its diameter is larger than the ligand/QD interface thickness where the surface dipole layer arises (Supplementary Fig. 5), $\Delta E$ is proportional to the number of ligands, $N$, and the effective surface dipole created by an adsorbed ligand, $\tilde{p}$, and it is inversely proportional to the surface area; $\Delta E \propto N\tilde{p}/r^2$, where $r$ is the QD radius. The effective surface dipole, $\tilde{p}$, is determined by both the intrinsic dipole of the free ligand, calculated in vacuum and the induced dipole at the ligand/QD interface (surface dipole);[7,14] we assumed that the former is proportional to the projected ligand dipole, $p$, and that the latter is the same for all of the R-$CA^-$ ligands since the ligand binding environment (bidentate carboxylate) remains constant throughout the library. The computed band edge shifts were then fit with the linear equation $E = E_0 + \Delta E = E_0 + A(N/r^2)p$, where $E_0$ and $A(N/r^2)$ were considered as variables. Using the experimental QD diameter of 3.2 nm ($r = 1.6$ nm) and $N = 100$ ligands (determined via quantitative NMR spectroscopy), $A(N/r^2)$ varies between 0.255 and 0.355 eV per Debye across the various models (Supplementary Table 4), which are of the same order of magnitude as those determined experimentally (Fig. 4a).

**Surface chemistry effects impacting QD band edge energies.** Despite the good agreement between experiment and computation, in our computational modeling we do not observe any significant deviation between fluorinated and non-fluorinated ligand/QD band edge data sets, which was clearly observed in our experimental measurements. To explain this deviation, we explored various surface-related phenomena. It is conceivable that the number of ligands coordinating the QD surface may vary across the ligand set due to differences in steric-mediated packing densities. Recently, we utilized a combination of NMR spectroscopy and spectrophotometric absorption titration to monitor and characterize the solution-phase ligand exchange of native $OA^-$ ligands for 2,6-F-$CA^-$ ligands at PbS QD surfaces. We found that the total number of coordinating carboxylate ligands remains constant during these X-type ligand exchanges, demonstrating a stoichiometric (1:1), self-limiting exchange reaction that likely proceeds through a proton transfer between native $OA^-$ and R-CAH free in solution[19,23]. Here, we demonstrate that the ligand exchange is driven to completion, as evidenced by FTIR and $^1H$ NMR spectroscopic analyses; thus, each removed OAH is replaced by one R-$CA^-$. In agreement, XPS elemental analysis shows that the C:Pb ratio (Supplementary

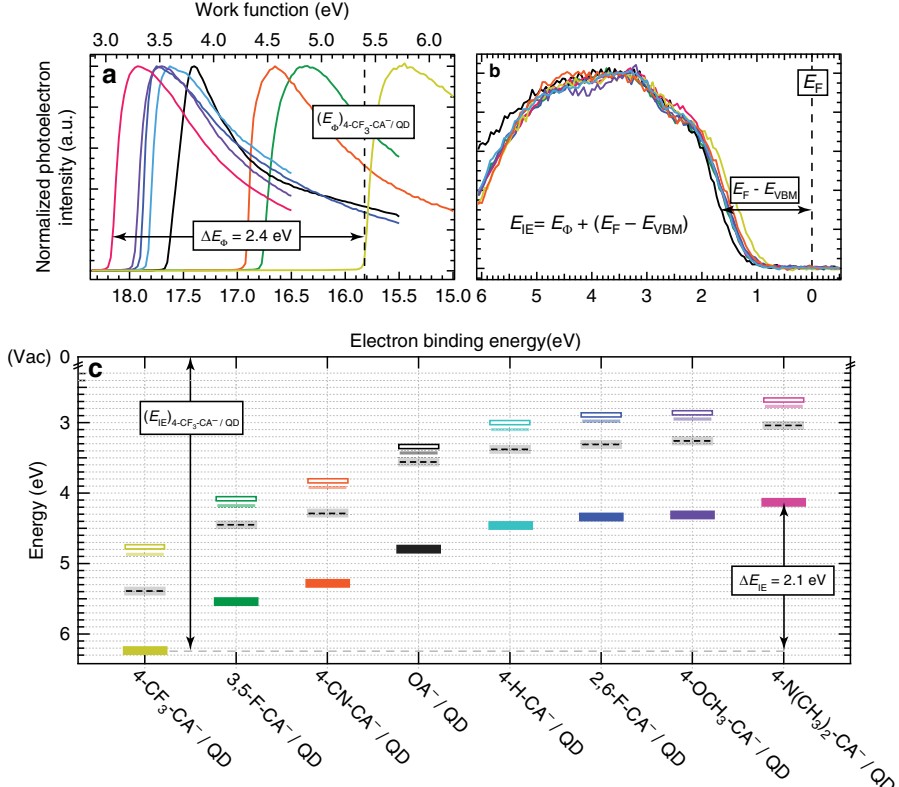

**Figure 3 | Photoelectron spectroscopy measurements of ligand/QD complexes.** (**a**) Secondary electron cut-off region of XPS spectra used to determine ligand/QD film work function. (**b**) Valence band edge region of XPS spectra used to extract the ligand/QD film VBM with respect to the Fermi energy $(E_F - E_{VBM})$. The VBM energies were extracted as described in Supplementary Note 3. The dashed black line represents the instrument equilibrated Fermi energy at an electron binding energy of 0 eV. (**c**) Band edge energies of films fabricated from OA$^-$ and R-CA$^-$ terminated 3.2 nm diameter PbS QDs; ionization energy (solid rectangles) and work function (dashed black lines). We also include the conduction band minimum ($E_{CBM}$, open rectangles) that is determined from summing the optical gap determined from absorbance measurements (opaque lines) and calculated exciton binding energy. The variation in the measurements made for duplicate samples was lower than the instrumental noise; therefore, the uncertainty of the ionization energy and work function values are represented as the vertical height of the closed rectangle and surrounding light gray rectangle, respectively.

Table 2), once normalized for ligand stoichiometry, remains nominally constant between the OA$^-$/QDs and R-CA$^-$/QDs indicating the ligand-to-QD ratio remains constant. Next we considered that, in the calculations, the ligands are oriented mostly perpendicular to the QD surface, while experimental configurations may vary, for example, due to steric or cooperative packing at the surface. However, all three of our models with different ligand binding geometries/orientations showed similar behaviour once renormalized for ligand surface coverage (Fig. 4b).

Another possible explanation for the difference in behaviour between the fluorinated and non-fluorinated R-CA$^-$ ligands is variability in ligand/ligand electrostatic interaction. Strong intra-QD and inter-QD ligand interactions may result in reduced effective ligand dipole moment due to dipolar screening/depolarization fields[24]. We modelled intra-QD ligand effects within a single-QD ligand shell by re-calculating projected ligand dipoles taking into account ligand polarizability effects (see Supplementary Note 7 for computational details). We found that the absolute magnitude of the renormalized effective ligand dipoles as a function of ligand functional group changes substantially, but the overall trend does not (Supplementary Fig. 6). In addition, we performed slab simulations and, again, found that the effective dipole moment is reduced in close packed ligand monolayers, but the original trend is retained (Supplementary Fig. 7). We thus conclude that intra-QD ligand surface polarization is unlikely to be the effect responsible for the observed differences.

Finally, we explored inter-QD ligand electrostatic effects by generating structural models of a QD film assembled from Model B by reducing the distance between periodic replicas in a simple cubic lattice in directions $x$ and $y$ (Fig. 4d,e). The QD film IE was computed by determining the HOMO (highest occupied molecular orbital) of the monolayers relative to the average electrostatic potential in the middle of the vacuum region along the $z$ direction. Figure 4c shows the change in HOMO position when isolated QDs (closed squares) are brought together to form a monolayer of QDs with inter-digitized ligand shells (open squares). We expect that inter-digitization reduces the effective dipole moment felt by the QD core since the dipole moment of the ligands of neighbour QDs point in a direction opposing the ligands of the QD in question. Indeed, we find that inter-digitization reduces the magnitude of the band edge shift in the monolayer, and we expect that such ligand inter-digitization-dependent renormalization effects would be even more pronounced in three-dimensional films.

Ligand shell inter-digitization is commonly observed for QD films fabricated using aliphatic, monodentate ligands[25,26], and likely occurs with R-CA$^-$ ligands. However, we hypothesize that the extent of inter-QD ligand shell inter-digitization is mediated by the functional group on the aromatic ring. Specifically, QD ligand shells composed of fluorinated cinnamates likely do not inter-digitate due to dipolar fluorine–fluorine electrostatic interactions and, thus, will exhibit band edge positions closer to those of the isolated QDs (Fig. 4c, closed squares). Fluorine is

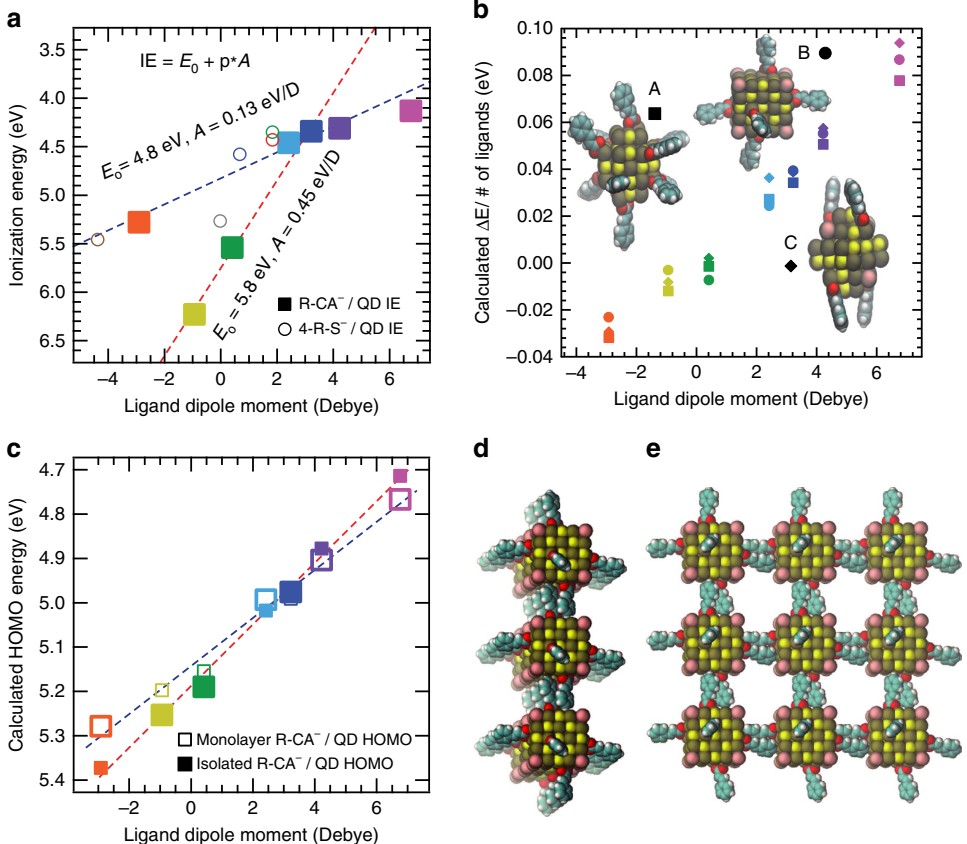

**Figure 4 | Band edge shifts of ligand/QD complexes.** (**a**) Experimentally measured R-CA⁻/QD (solid squares) and 4-R-S⁻/QD (open circles) ionization energies as a function of calculated ligand dipole. The variation of ionization energy measurements made for duplicate samples was lower than the instrumental noise; therefore, the uncertainty of the data is less than the width of the data markers. The brown and grey open circles are data taken from Bent *et al.*[11] that correspond to 4-NO₂-S⁻ and 4-F-S⁻ capped PbS QDs, respectively. The blue dashed line is a fit to the data that includes the points associated with non-fluorinated ligands, and the red dashed line is a fit to the data that only includes the points associated with fluorinated ligands. (**b**) Calculated band edges renormalized by the number of ligands as computed for the three different isolated ligand/QD structural models with varying surface coverage as a function of the projected ligand dipole. (**c**) The energy of the highest occupied molecular orbital for isolated QDs (filled squares) and the square lattice of monolayer of QDs (open squares). The lines are guides to the eye and the larger data points represent the proposed physically accurate QD film environment—either isolated (closed squares) or inter-digitized monolayer (open squares)—for each R-CA⁻/QD sample based on inter-QD ligand shell electrostatic arguments as described in the text. (**d**) Side and (**e**) top view of a 3 × 3 repetition of the unit cell for an example R-CA⁻/QD monolayer.

the most electronegative element on the periodic table, therefore, C–F bonds are highly dipolar leaving the fluorine (carbon) atom with a slight negative (positive) charge. Thus, the fluorine atoms of interacting ligand shells could repel one another and prevent inter-QD ligand shell inter-digitization. A simple physical model of ligand shell inter-digitization is shown in Supplementary Fig. 8. Using the principles of inter-QD ligand electrostatic effects mediating ligand shell inter-digitization, we find good agreement between the computational predictions and the experimental trends (see the large square data markers in Fig. 4c, and compare to Fig. 4a). The fluorinated ligand/QD complexes are shifted to deeper IEs because their ligand shells remain isolated (Fig. 4c, large, closed squares) compared to the non-fluorinated ligand/QD complexes, which undergo ligand shell inter-digitization induced dipole moment renormalization (Fig. 4c, large, open squares).

## Discussion

Through a combination of experiment and *ab initio* calculations, we established that ligand dipole is a critical parameter for predicting the overall direction and upper limit to the magnitude of band edge shifting in PbS QD thin films; however,

ligand–ligand electrostatic effects that dictate inter-QD ligand inter-digitization and effectively screen the ligand shell dipole must also be considered. Controlling absolute band edge positions is an important design criterion for a large variety of potential solid-state and colloidal QD applications. We found that IE (Φ) of our ligand/QD complex library ranges from 6.2 (5.4) eV for the 4-CF₃-CA⁻/QD complex to 4.1 (3.0) eV for the 4-N(CH₃)₂-CA⁻/QD complex. For comparison, we fabricated other PbS ligand/QD thin films that have been studied previously, following layer-by-layer film fabrication procedures as specified in the literature using a similar PbS QD system[14,16,17]. We found good agreement between the XPS band edge values we measured and those reported elsewhere, giving us confidence that our R-CA⁻/QD complex library induces the deepest and shallowest absolute band edges for PbS QD thin films, to date (Supplementary Fig. 9).

The procedures developed here are expected to translate to other R-CA⁻ ligands, of which there are greater than 500 known substitutional groups. In addition, we found similar trends of band edge shifts for a second class of ligands, functionalized benzenethiolates, including a similar deviation between theoretical and experimental ordering for fluorinated and non-fluorinated ligands. Our results can be extrapolated to larger

QDs using an electrostatic model, showing that band edge shifts should be independent of QD diameter if the ligand surface coverage (ligands per nm$^2$) remains constant as a function of size.

We also demonstrated a simple, robust and scalable solution-phase ligand exchange method that yielded well-defined, highly tunable chemical systems allowing us to cleanly and systematically correlate QD film band edge shifts with surface chemistry. Solution-phase ligand exchanges will benefit numerous synthetic processes. For example, conventional methods to fabricate QD thin films employ solid-state layer-by-layer, batch-processing fabrication techniques because the ligands most commonly employed do not impart QD colloidal stability. Here we deposited electronically coupled films via a single-deposition step from low boiling point solvents. Single-step deposition from a colloidal QD 'ink' allows for high-throughput, roll-to-roll processing where film thickness and morphology can be controlled through QD concentration and solvent composition[27]. Furthermore, single-step deposition techniques allow for QD superlattice formation, which holds the potential for highly conductive QD films through band-like transport[28]. The flexibility of employing organic ligands to produce functional inorganic/organic systems is very attractive for many emerging applications, and we expect the trends established here relating ligand properties with ligand/QD electronic properties to serve as a guide for further studies using *a priori* approaches. While not the subject of the present work, we find that in addition to tuning band edge positions, all of the R-CA$^-$ ligands induce broadband optical absorbance enhancement demonstrating that the beneficial band edge tuning does not come at the expense of degraded optical properties.

## Methods

**General methods.** All manipulations were performed using standard air-free techniques on a Schlenk line under nitrogen atmosphere or in a nitrogen-filled glovebox unless otherwise indicated.

**Materials.** All chemicals were used as received without further purification unless noted. Anhydrous octane (≥99%), anhydrous diethylene glycol dimethyl ether (diglyme, 99.5%), N,N′-diphenylthiourea (98%), anhydrous toluene (99.5%), anhydrous tetrachloroethylene (≥99.9%), anhydrous methyl acetate (MeOAc, 99%), anhydrous 1,2-dichlorobenzene (99%), anhydrous hexane (≥99%), anhydrous dichloromethane (DCM, ≥99.8%), anhydrous acetonitrile (99.8%), anhydrous isopropanol (99.5%), anhydrous chloroform-*d* (CDCl$_3$, ≥99.8%), acetone (≥99.9%, degassed), anhydrous tetrahydrofuran (≥99.9%), 1,1,1,3,3,3-hexafluoro-2-propanol (≥99%, degassed), *trans*-cinnamic acid (4-H-CAH, ≥99%), *trans*-2,6-difluorocinnamic acid (2,6-F-CAH, 99%), *trans*-3,5-difluorocinnamic acid (3,5-F-CAH, 99%), *trans*-4-(trifluoromethyl)cinnamic acid (4-CF$_3$-CAH, 99%), 4-methoxycinnamic acid (4-OCH$_3$-CAH, 99%), 4-(dimethylamino)cinnamic acid, predominantly *trans* (4-N(CH$_3$)$_2$-CAH, 99%), ferrocene (Cp$_2$Fe, 98%), triethylamine (≥99%), benzenethiol (4-H-SH, ≥98%), 4-aminobenzenethiol (4-NH$_2$-SH, 97%), 4-methylbenzenethiol (4-CH$_3$-SH, 98%), 4-bromobenzonitrile (99%), anhydrous acrylic acid (99%), N,N-dicyclohexylmethylamine (97%) and anhydrous magnesium sulfate (MgSO$_4$) were obtained from Sigma Aldrich. 4-(trifluoromethyl)benzenethiol (4-CF$_3$-SH, 97%) was obtained from Alfa Aesar. Bis(tri-*tert*-butylphosphine)palladium (98%) was obtained from Strem Chemicals. Hydrochloric acid (HCl, ACS) was obtained from Macron Fine Chemicals.

**Oleate capped PbS QD synthesis.** Oleate capped PbS QDs with a core diameter of 3.2 nm corresponding to a first exciton transition energy centred at ∼1.3 eV were synthesized following the substituted thiourea protocol developed by Hendricks *et al.*[18] First, hydroxide-free Pb(oleate)$_2$ was prepared and purified. In a nitrogen glovebox, 8.81 g Pb(oleate)$_2$ and 150 ml anhydrous octane were added to a 2-neck 250 ml Schlenk flask equipped with a magnetic stir bar and sealed using a glass stopcock and two rubber septa. Separately, 1.74 g of N,N′-diphenylthiourea and 5 ml of diglyme were mixed in a 20 ml scintillation vial and sealed with a rubber septa. After transferring to a Schlenk line, both vessels were brought to 95 °C in an oil bath under nitrogen and allowed to stir for ∼30 min or until both solutions were clear. Subsequently, the N,N′-diphenylthiourea diglyme solution was quickly injected into the Pb(oleate)$_2$ octane solution under vigorous stirring. After 60 s, the flask, now containing a dark brown solution, was removed from the oil bath and allowed to cool to room temperature. The septa were then removed under positive nitrogen pressure and replaced with glass stoppers so the volatiles could be removed from the flask under vacuum. The flask was transferred to a nitrogen-filled glovebox and the sticky, brown reaction crude was dispersed in ∼40 ml toluene and split between four 50 ml centrifuge tubes and centrifuged at 7,000 r.p.m. for 10 min. The brown nanocrystal solution was decanted into four new centrifuge tubes and the remaining dark pellets were discarded. To each centrifuge tube, ∼30 ml of methyl acetate was added to precipitate the QDs and then centrifuged at 7,000 r.p.m. for 10 min. This cycle of PCR purification using toluene and methyl acetate was repeated a total of three times. The QD product was dried under vacuum and finally suspended in hexane for storage in a nitrogen-filled glovebox. Due to the large yield (multi-gram scale) of the QD synthesis, we performed all experiments on the same stock QD sample, thereby eliminating the effects of sample-to-sample variations.

**Ligand exchange and PCR purification.** In a nitrogen-filled glovebox, a QD DCM solution was prepared and its concentration determined using absorbance spectroscopy[29]. The amount of ligand necessary to completely exchange the standardized QD solution (∼800–900 ligands per QD) was dissolved in its compatible solvent mixture in a 20 ml scintillation vial equipped with a magnetic stir bar. The QD solution volume:ligand solution volume was 6:1–10:1 (some of the ligand solution components used in this study are known to precipitate QDs out of solution[9], but this was not observed at these ratios). The ligand solution was added dropwise to the QD solution with vigorous stirring, and the exchange was allowed to proceed for ∼10 min. The exchanged QDs were isolated from byproducts (OAH) and excess R-CAH via multiple precipitation, centrifugation and redissolution (PCR) cycles (isolation parameters for specific ligands detailed in Supplementary Table 1).

**Thin film fabrication.** PbS ligand/QD thin films were fabricated via a single-deposition step using spin-coating. In a nitrogen-filled glovebox, PbS ligand/QD solutions at concentrations of ∼200 mg ml$^{-1}$ in the solvents indicated in Supplementary Table 1 were prepared. The solutions were dispensed onto cleaned substrates through a 0.2 μm PTFE syringe filter to remove any aggregated particles and spun at 1,500 r.p.m. for 30 s. The substrates were finally annealed at 90 °C for 20 min, which produced QD films that were sufficiently thick that XPS measurements probed outside of the band-bending region of the Au substrate/QD thin film interface and uniform/devoid of pinholes seeing as Au was not detected during XPS measurements.

**Ultraviolet–visible–near infrared absorbance spectroscopy.** Ground-state optical absorbance spectra were recorded using a Cary 500 UV–vis–NIR spectrophotometer.

**FTIR spectroscopy.** FTIR absorbance measurements were taken on a Thermo-Nicolet 6700 FTIR spectrometer in transmission mode with a resolution of 4 cm$^{-1}$. Clean Si plates were used for background measurements, and films of oleate capped QDs were drop cast onto the Si plates from hexane. Films of ligand-exchanged QDs were cast from solvents detailed in Supplementary Table 1. Spectra with sloping baselines were baseline-corrected.

**$^1$H NMR spectroscopy.** $^1$H NMR spectra were recorded on a Bruker Avance III 400 MHz instrument and acquired with sufficiently long delay to allow complete relaxation between pulses (30 s). $^1$H NMR spectroscopy was also performed under quantitative conditions to determine the number of OA$^-$ ligands bound to the QD core for the as prepared sample. First, we determined the concentration of the stock QD sample using ultraviolet–visible–near infrared absorption spectroscopy. Then, we added a known amount of ferrocene (10 H's) to the QD sample as an internal standard, which allowed us to estimate the total number of surface bound oleate ligands using the well-resolved vinyl proton peak (H$^V$, 2 H's). We find that there are ∼100 ligands per PbS QD, giving an estimated OA$^-$ surface grafting density of 3.1 ligands per nm$^2$.

**X-ray photoelectron spectroscopy.** Thin films of Au (100 nm) on Cr (10nm) on glass were thoroughly cleaned and used as substrates for XPS measurements. XPS measurements were performed on a Physical Electronics, Inc. 5600 ESCA instrument, which has been discussed in detail previously[30]. Briefly, the radiation is produced by a monochromatic Al (Kα) source centred at 1486.6 eV. The valence band spectra were taken with a step size of 0.05 eV and a pass energy of 5.85 eV. The electron binding energy scale was calibrated using the Fermi edge of cleaned metallic substrates (Au, Mo, Cu and/or Ag), giving the spectra an uncertainty of ±0.05 eV. None of the ligand/QD samples studied here showed significant photocharging during the XPS measurements.

**General considerations for DFT calculations.** Calculations were performed using the plane wave basis set code Quantum-ESPRESSO[31] using the PBE parametrization of the generalized gradient approximation exchange-correlation functional[32], optimized norm-conserving pseudopotentials[33,34] and a wave function cut-off of 80 and 60 Ry for structural relaxations and single-point calculations, respectively. Absolute single-particle energies of the isolated QDs were computed with respect to the vacuum level by determining the average electrostatic potential at the cell

boundary and again by applying Makov–Payne-like corrections. See Supplementary Notes 1, 4–6, and Supplementary Figs 3–7 for the details on the construction of the structural models and the analysis of the results.

**Data availability.** The authors declare that all data supporting this work are contained in graphics displayed in the main text or in supplemental information. Data used to generate these graphics are available from the authors on request.

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

## Acknowledgements

We would like to thank Greg Pach and Boris Chernomordik for help with thin film deposition, and Nick Anderson for helpful discussions. D.M.K., E.M.M., J.G. and M.C.B. respectfully acknowledge support through the Division of Chemical Sciences, Geosciences and Biosciences, Office of Basic Energy Sciences, Office of Science within the US Department of Energy for the ligand exchange and characterization studies through contract No. DE-AC36-08GO28308 from DOE. N.P.B. and G.G. were supported as part of the Center for Advanced Solar Photophysics, an Energy Frontier Research Center funded by the Office of Basic Energy Sciences, Office of Science with DOE. M.V. was supported by Laboratory Directed Research and Development (LDRD) funding from Argonne National Laboratory, provided by the Director, Office of Science, of the U.S. Department of Energy under Contract No. DE-AC02-06CH11357. This research used computational resources of: the National Energy Research Scientific Computing Center, a DOE Office of Science User Facility supported by the Office of Science of the U.S. DOE under Contract No. DE-AC02-05CH11231; and the Los Alamos National Laboratory Institutional Computing Program, which is supported by the U.S. Department of Energy National Nuclear Security Administration under Contract No. DE-AC52-06NA25396. A.S. was supported through Colorado School of Mines start-up funds. B.W.M. respectfully acknowledges the sponsorship and support of the United States Air Force Institute of Technology.

## Author contributions

D.M.K., M.C.B. and A.S. conceived the original ideas and designed the experiment; D.M.K. carried out the ligand exchange experiments, characterized the ligand/QD complexes, and analyzed the experimental data; E.M.M. performed the XPS measurements and analyzed the data. B.W.M. and A.S. designed, synthesized and characterized select cinnamic acid derivatives; J.G. characterized select cinnamic acid derivatives and ligand/QD complexes; M.V., N.P.B. and G.G. designed the computational study. M.V. and N.P.B. carried out the calculations and analyzed the data. D.M.K., M.V., G.G. and M.C.B. wrote the manuscript. A.J.N. assisted in the preparation of the manuscript. All authors discussed the results and commented on the manuscript.

## Additional information

**Competing interests:** The authors declare no competing financial interests.

