## [Peer Review File · Nature Communications]

Reviewers' comments:

Reviewer #1 (Remarks to the Author):

This is an extensive, detailed, and clean study of the relationship between ligand functionalization on one type of colloidal nanocrystal, PbS, and the energies of its valence and conduction-band edges. It shows the dramatic effects of surface dipoles on the electronic structure of very small particles, and explains the often-seen deviation of fluorinated ligands from the trends of their hydrocarbon analogs. I think the hypothesis of different degrees of dipole screening due to inhibited intercalation of fluorinated ligands is a reasonable one. It's a very useful paper and is appropriate for Nature Communications. I have only a few suggestions:

1) in the intro, the authors say: "These studies clearly demonstrated that modifying the ligand/QD 55 interfaces produces quite distinct chemical systems; yet clear relationships between QD surface chemistry and the resulting optoelectronic properties of the ligand/QD hybrid organic/inorganic systems have not been clearly established." This statement suggests to the reader that the authors will clearly establish these relationships. The authors do so, but for a specific parameter of the ligands (dipole moment) for a specific type of NC (PbS), and a specific property (band-edge structure). The authors should be very clear in the intro that this is what they will deliver. They hold the binding group constant, so they are not exploring the influence of electronic conjugation between the NC core and the ligand, for example.

2) page 2-3, line 77-78: carboxylates are "hard" bases and Pb^{2+} is a "borderline" acid, so statement that "the vinyl linkage of R-CAH allows for electronic coupling of the dipole active portion of the ligand to the QD core (e.g. through orbital hybridization)" is a bit misleading. Yes, every direct orbital-orbital interaction has some trivial amount of mixing, but there, to my knowledge, is not evidence to suggest that carboxylate and lead orbitals hybridize to any significant extent, at least not the way that sulfur groups hybridize with Cd, for instance. The influence of a dipole moment on the energy levels of the QD core are probably complicated (there are resonance versus inductive effects, for instance), and I don't think the authors distinguish between the two and therefore don't clarify the role of orbital conjugation in this whole picture. I think saying that the binding group provides some electronic coupling between the NC and the ligand is sufficient.

3) page 3: the statement about conjugated carboxylates enhancing optical absorption is not supported by all of the references listed. I believe Refs 10,11 refer to so-called "exciton delocalizing ligands", namely thiolates. The statement is also true for dithiocarbamates, but, not carboxylates based on those refs. The authors should not say that binding of all short-chain aromatics causes this enhancement.

Reviewer #3 (Remarks to the Author):

After careful review of the revised manuscript and the response to comments and critique raised in the initial review of the manuscript, questions about the level of novelty of the present work remain. The quality of the work is high and the rigorous quantitative correlation between ligand dipole and band edge position is certainly important. Moreover, the revised manuscript provides a clearer scope, but this was also done at the expense of taking out elements of the study that didn't coherently fit but are

instead deferred to separate papers. Despite the quality of the work, the basis of novelty seems to narrow for recommended publication as a nature communication.

Response to Reviewers:

Reviewer #1 (Remarks to the Author):

This is an extensive, detailed, and clean study of the relationship between ligand functionalization on one type of colloidal nanocrystal, PbS, and the energies of its valence and conduction-band edges. It shows the dramatic effects of surface dipoles on the electronic structure of very small particles, and explains the often-seen deviation of fluorinated ligands from the trends of their hydrocarbon analogs. I think the hypothesis of different degrees of dipole screening due to inhibited intercalation of fluorinated ligands is a reasonable one. It's a very useful paper and is appropriate for Nature Communications. I have only a few suggestions:

Response: We thank the reviewer for the careful reading of our revised manuscript and for the suggestions.

1) in the intro, the authors say: "These studies clearly demonstrated that modifying the ligand/QD interfaces produces quite distinct chemical systems; yet clear relationships between QD surface chemistry and the resulting optoelectronic properties of the ligand/QD hybrid organic/inorganic systems have not been clearly established." This statement suggests to the reader that the authors will clearly establish these relationships. The authors do so, but for a specific parameter of the ligands (dipole moment) for a specific type of NC (PbS), and a specific property (band-edge structure). The authors should be very clear in the intro that this is what they will deliver. They hold the binding group constant, so they are not exploring the influence of electronic conjugation between the NC core and the ligand, for example.

Response: We thank the reviewer for pointing this out and have revised the introduction of the manuscript to more clearly reflect that we are correlating surface chemistry physicochemical properties, specifically ligand dipole and ligand shell inter-digitization, with experimentally measured band edge shifts, specifically for PbS QD thin films.

2) page 2-3, line 77-78: carboxylates are "hard" bases and Pb^{2+} is a "borderline" acid, so statement that "the vinyl linkage of R-CAH allows for electronic coupling of the dipole active portion of the ligand to the QD core (e.g. through orbital hybridization)" is a bit misleading. Yes, every direct orbital-orbital interaction has some trivial amount of mixing, but there, to my knowledge, is not evidence to suggest that carboxylate and lead orbitals hybridize to any significant extent, at least not the way that sulfur groups hybridize with Cd, for instance. The influence of a dipole moment on the energy levels of the QD core are probably complicated (there are resonance versus inductive effects, for instance), and I don't think the authors distinguish between the two and therefore don't clarify the role of orbital conjugation in this whole picture. I think saying that the binding group provides some electronic coupling between the NC and the ligand is sufficient.

Response: We agree with the reviewer that there is no direct evidence (experimental or computational) presented here for orbital hybridization between carboxylate and lead; therefore, we have removed all references to orbital hybridization between the QD organic ligand shell and inorganic core. We have revised the above mentioned passage to read, "the vinyl linkage of R-CAH allows for electronic coupling of the dipole active portion of the ligand to the QD core."

3) page 3: the statement about conjugated carboxylates enhancing optical absorption is not supported by all of the references listed. I believe Refs 10,11 refer to so-called "exciton delocalizing ligands", namely thiolates. The statement is also true for dithiocarbamates, but, not carboxylates based on those refs. The authors should not say that binding of all short-chain aromatics causes this enhancement.

Response: We included refs. 10 and 11 here because, to the best of our knowledge, they were the first studies that reported broadband optical absorbance enhancement arising from ligand/QD interactions. It is true that they used a different ligand system, namely benzenethiolates, which could have a different mechanism for inducing broadband optical absorbance enhancement compared with cinnamates; therefore, we have removed

refs. 10 and 11 as support for the above mentioned claim, and leave the reference that specifically details broadband optical absorbance enhancement in the carboxylate ligand platform.

Reviewer #3 (Remarks to the Author):

After careful review of the revised manuscript and the response to comments and critique raised in the initial review of the manuscript, questions about the level of novelty of the present work remain. The quality of the work is high and the rigorous quantitative correlation between ligand dipole and band edge position is certainly important. Moreover, the revised manuscript provides a clearer scope, but this was also done at the expense of taking out elements of the study that didn't coherently fit but are instead deferred to separate papers. Despite the quality of the work, the basis of novelty seems to narrow for recommended publication as a nature communication.

Response: First, we would like to thank the reviewer for the careful reading of our revised manuscript and for their response. However, we disagree with the reviewer that this work is not sufficiently novel for Nature Communications. In this work, we establish a novel, yet simple, robust, and scalable solution-phase ligand exchange procedure that results in an extremely clean and well defined QD surface chemistry (contrasting the previously reported solid-state ligand exchange procedures detailed in the text's introduction). Subsequently, using facile single-step deposition techniques, we fabricate electronically conductive thin films of PbS QDs that exhibit the deepest and shallowest band edge positions reported in the literature spanning an unprecedented range of over 2.0 eV. Our work is the first to quantitatively establish that ligand dipole moment can be a predictor of the magnitude and direction of QD band edge shifts, but we also find that other electrostatic effects are critical parameters to consider (specifically, ligand surface grafting density, binding orientation, and, most importantly, ligand shell inter-digitization). The absolute band edge position of semiconductors is a critical design criterion for a large variety of potential optoelectronic applications, and our clear control of QD band edge position coupled with solution processability will undoubtedly appeal to a large audience.

We agree with the reviewer that the revisions to the previous manuscript provide a clearer scope for the current manuscript, but disagree that removing certain elements from the previous manuscript detract from the current manuscript in any way. In fact, we find this comment somewhat contradictory to their previous round of comments where they recommended we reorganize our previous manuscript for clarity. If anything, removing specific elements allowed us to simplify, clarify, and better organize the main message of the work presented here, as they originally suggested.

Major Changes to Manuscript:

- Page 2, lines 15-21: Replaced text for clarity based on the points made by Review #1 in comment 1.

“These studies demonstrated that modifying the ligand/PbS QD interface produces quite distinct chemical systems, yet while previous studies of PbS QD films fabricated *via* solid-state ligand exchange have attempted to relate QD band edge energy shifts to ligand dipole moment, a quantitative relationship between ligand dipole moment and band edge shift was never reported.”

to

“All of these studies demonstrated that modifying the ligand/PbS QD interface produces quite distinct chemical systems, and some even suggested a link between QD band edge energy shifts and ligand dipole moment; however, due to the uncontrolled and ill-defined physicochemical nature of solid-state ligand exchanges, a clear and quantitative relationship has never been reported.”

- Page 2, lines 27-30: Removed text similar to that which was used to replace the above text.

“While previous studies of PbS QD films fabricated via solid-state ligand exchange have attempted to relate QD band edge energy shifts to ligand dipole moment, a quantitative relationship between ligand dipole moment and band edge shift was never reported.”

- Page 3, line 6-7: Removed text to address point made by reviewer 1 in comment 2.

“(e.g. through orbital hybridization),”

- Page 3, line 9: Removed text to address point made by reviewer 1 in comment 3.

“of short chain, aromatic ligands such as”

- Page 3, line 9: Removed references 10 and 11 to address point made by reviewer 1 in comment 3.
- Page 10, line 40: Replaced text for clarity based on the points made by Review #1 in comment 1.

“systematically correlate ligand/QD electronic properties with QD surface chemistry”

to

“systematically correlate QD band edge shifts with surface chemistry”